# *Candida albicans* Promotes the Antimicrobial Tolerance of *Escherichia coli* in a Cross-Kingdom Dual-Species Biofilm

**DOI:** 10.3390/microorganisms10112179

**Published:** 2022-11-03

**Authors:** Shintaro Eshima, Sanae Kurakado, Yasuhiko Matsumoto, Takayuki Kudo, Takashi Sugita

**Affiliations:** 1Department of Microbiology, Meiji Pharmaceutical University, Kiyose 204-8588, Tokyo, Japan; 2Department of Pharmacy, Toshiba Rinkan Hospital, Sagamihara 252-0385, Kanagawa, Japan; 3Department of Clinical Laboratory, Toshiba Rinkan Hospital, Sagamihara 252-0385, Kanagawa, Japan

**Keywords:** antibacterial tolerance, biofilm, *Candida albicans*, *Escherichia coli*, meropenem

## Abstract

Cross-kingdom multi-species biofilms consisting of fungi and bacteria are often resistant to antimicrobial treatment, leading to persistent infections. We evaluated whether the presence of *Candida albicans* affects the antibacterial tolerance of *Escherichia coli* in dual-species biofilms and explored the underlying mechanism. We found that the survival of *E. coli* in the presence of antibacterial drugs was higher in dual-species biofilms compared to single-species biofilms. This tolerance-inducing effect was observed in *E. coli* biofilms that were treated with a *C. albicans* culture supernatant. To explore the antibacterial tolerance-inducing factor contained in the culture supernatant and identify the tolerance mechanism, a heated supernatant, a supernatant treated with lyticase, DNase, and proteinase K, or a supernatant added to a drug efflux pump inhibitor were used. However, the tolerance-inducing activity was not lost, indicating the existence of some other mechanisms. Ultrafiltration revealed that the material responsible for tolerance-inducing activity was <10 kDa in size. This factor has not yet been identified and needs further studies to understand the mechanisms of action of this small molecule precisely. Nevertheless, we provide experimental evidence that *Candida* culture supernatant induces *E. coli* antibacterial tolerance in biofilms. These findings will guide the development of new treatments for dual-species biofilm infections.

## 1. Introduction

Biofilms on the surfaces of medical devices, such as central venous catheters (CVCs), are often resistant to antimicrobial agents [1]. These biofilms are associated with approximately 80% of all infections, which often become chronic due to a recurrence rate of approximately 65–80% [2,3]. Patients receiving parenteral nutrition therapy using CVCs are more likely to develop catheter-related bloodstream infections, where 15–37% of these infections are polymicrobial [4,5].

Surgical removal of infected medical devices is the basic treatment for biofilm infections; however, this severely burdens the patients. Many researchers have sought medical treatments for biofilm infections; however, it is extremely difficult to eliminate biofilms that feature multiple species. Most biofilms contain several microbial species and are particularly difficult to treat when the infection is caused by species from different kingdoms [6]. Both antibacterial and antifungal agents are often required; however, microbes may tolerate these drugs [7].

*Candida* inhabits the intestinal tract as well as the oral mucosa and can trigger bloodstream and organ infections in the immunocompromised host [8]. The number of *Candida* infections has been steadily increasing each year due to the increase in the number of immunocompromised hosts [9]. Studies have shown that *C. albicans* is the predominant causative agent of catheter-related bloodstream infections, followed by *Staphylococcus* and *Enterococcus* species, and 25% of patients with Candidemia have polymicrobial infections [10,11].

*Escherichia coli*, a facultative anaerobic Gram-negative bacillus, causes urinary tract infections (UTIs) and bloodstream infections (BSIs) after translocation from the intestine to the urinary tract or bloodstream, respectively [12]. Over the past decade, the incidence of BSIs caused by *E*. *coli* has increased worldwide, along with the mortality rate, which is between 5% and 30% [13,14,15]. Research has demonstrated that some *E*. *coli* isolates from patients with UTIs and BSIs form biofilms [16,17].

Since we hypothesized that *E. coli* clinically isolated from a Japanese hospital might contain biofilm-forming strains, the biofilm-forming capacity of *E. coli* isolates from the blood samples of patients was tested. Identifying compounds that are effective against drug-resistant bacteria and compounds that reduce drug resistance [18,19] and research on drug resistance caused by microbial components [20,21,22] were reported. We also hypothesized that *C. albicans* might produce substances that affect drug resistance in biofilm-forming *E. coli*. We found that dual-species biofilms formed by *E*. *coli* and *C*. *albicans* were tolerant to antibacterial drugs, possibly due to a tolerance-inducing factor present in the culture supernatant of the *Candida* biofilms.

## 2. Materials and Methods

### 2.1. Escherichia coli Isolates

Fifty-two *E*. *coli* isolates were obtained from the blood cultures of patients treated at Toshiba Rinkan Hospital, Kanagawa, Japan, between 2013 and 2020. The primary diseases are shown in Figure 1. The isolates were initially identified using a MicroScan system (Auto SCAN4; Beckman Coulter, Brea, CA, USA) in conjunction with Neg combo panels. Identification was confirmed via 16S rRNA gene sequencing [23]. The microbial 16S rRNA gene was amplified using PCR with primers 27F (5′-GAGTTTGATCCTGGCTCAG-3′) and 1492R (5′-GGYTACCTTGTTACGACTT-3′) and was sequenced. Strains with >99% sequence similarity were defined as conspecific.

### 2.2. Biofilm Formation (E. coli Only)

Biofilms were created as described by de Brucker et al. [24] with some modifications. *E*. *coli* strains were grown in tryptic soy broth (Thermo Fisher Scientific, Waltham, MA, USA) at 37 °C for 24 h at 150 rpm. The pellets were collected via centrifugation and washed with phosphate-buffered saline (PBS). Cell suspensions were adjusted to an A_630_ value of 0.0001 in RPMI 1640 medium (Thermo Fisher Scientific) supplemented with 3-(*N*-morpholino) propanesulfonic acid (MOPS) (pH 7.3). Afterward, 100 µL cell suspension was seeded into each well of the flat-bottomed 96-well microtiter plates. After static incubation at 37 °C for 24 h, planktonic cells were removed, and adherent cells were washed three times with PBS. The biofilm formation capacity was evaluated using a crystal violet assay following the manufacturer’s instructions. Briefly, after the biofilms dried, 0.1% (*w*/*v*) aqueous crystal violet solution was added to each well and incubated at room temperature (10–30 °C) for 1 h. The staining solution was discarded, and the wells were washed three times with PBS. After the wells were dried, 33% (*v*/*v*) acetic acid was added, and the plates were incubated at room temperature (10–30 °C) for 30 min. Subsequently, absorbance at A_550_ was measured using a Microplate Reader (BIO-RAD, Hercules, CA, USA). Biofilm formation capacity was quantified as follows: high capacity, A_550_ > 0.8; low capacity, A_550_ 0.1–0.8; and no capacity, A_550_ < 0.1.

### 2.3. Drug Susceptibility Testing of Planktonic Cells and Biofilms (E. coli Only)

Four antibacterial agents were used for susceptibility testing: meropenem (MEPM, Fujifilm Wako Pure Chemical Corporation, Osaka, Japan), cefmetazole (CMZ, Tokyo Chemical Industry, Tokyo, Japan), ceftriaxone (CTRX, Tokyo Chemical Industry), and levofloxacin (LVFX; Nacalai Tesque, Kyoto, Japan). LVFX was dissolved in 1.4% acetic acid, while the other three agents were dissolved in dimethyl sulfoxide.

The drug susceptibilities of planktonic *E*. *coli* cells were determined using the MicroScan Neg combo panel (Beckman Coulter, Brea, CA, USA) following the Clinical and Laboratory Standards Institute (CLSI) M100-Ed31 protocol [25]. Antimicrobial agent ranges were 1 and 2 µg/mL for MEPM and CTRX, 8, 16 and 32 µg/mL for CMZ, and 0.5, 1, 2 and 4 µg/mL for LVFX. *E*. *coli* biofilms were prepared as described above. Antimicrobial agents (0, 12.5, 25 and 50 µg/mL for MEPM, CMZ, and CTRX, 0, 0.1, 1, 10 and 50 µg/mL for LVFX) were then added to the wells, and the plates were incubated statically at 37 °C for 24 h. The wells with 0 µg/mL drug contained an equivalent amount of each solvent and were used as drug-free controls. Afterward, planktonic cells were removed, and the wells were washed three times with PBS. Viable cells in the biofilms were counted using the colony-forming unit (CFU) method. Biofilm cells were resuspended in 100 µL PBS via vigorous pipetting. Next, the cell suspensions were diluted, plated onto nutrient agar and incubated. After 24–48 h, the CFUs were counted.

### 2.4. Drug Susceptibility Testing of Biofilms (C. albicans Only)

*C. albicans* SC5314 was grown on Sabouraud dextrose agar at 37 °C for 24 h. Cell suspensions were adjusted to an A_630_ value of 0.1 in RPMI 1640 medium supplemented with MOPS (pH 7.3). Standardized cell suspensions (100 µL) were seeded into each well of the flat-bottomed 96-well microtiter plates and incubated at 37 °C for 24 h. Planktonic cells were removed, and a medium with 50 μg/mL MEPM was added. The control wells contained an equivalent amount of DMSO. Thereafter, the plates were statically incubated at 37 °C for 24 h, the planktonic cells were removed, and the wells were washed three times with PBS. Viable cells in the biofilms were counted using the CFU method.

### 2.5. Drug Susceptibility Testing of Dual-Species Biofilms (E. coli and C. albicans)

*C. albicans* SC5314 was grown on Sabouraud dextrose agar at 27 °C for 24 h, and *E*. *coli* RB-3 was grown in tryptic soy broth (Thermo Fisher Scientific) at 37 °C for 24 h at 150 rpm. The *E. coli* pellets were collected via centrifugation and washed with PBS. The cell suspensions were adjusted to an A_630_ of 0.0001 and 0.1 for *E*. *coli* and *C*. *albicans*, respectively. The prepared cell suspensions (100 μL each) were mixed in the wells of flat-bottomed 96-well microtiter plates and incubated at 37 °C for 24 h. Planktonic cells were removed and a medium with 50 μg/mL MEPM was added. The control wells without antimicrobial agents contained an equivalent amount of DMSO. Thereafter, the plates were statically incubated at 37 °C for 24 h. The single-species biofilms also served as controls. The CFUs of *E*. *coli* and *C*. *albicans* were counted as described above. Approximately 1 μg/mL micafungin was added to the nutrient agar medium (to selectively grow *E*. *coli*) and 100 μg/mL streptomycin was added to the Sabouraud agar medium (to selectively grow *C*. *albicans*) as *E*. *coli* and *C*. *albicans* strains are susceptible to streptomycin and micafungin, respectively.

### 2.6. Effect of C. albicans Culture Supernatant on E. coli Biofilm Formation in the Presence of MEPM, CMZ, CTRX or LVFX

*C. albicans* biofilms were prepared as described above. After incubation at 37 °C for 24 h, the culture supernatant was collected and passed through a 0.22-µm pore membrane filter (Sartorius, Gottingen, Germany) to remove cells. *E. coli* biofilms were incubated with *Candida* supernatant and 50 μg/mL MEPM, CMZ, CTRX, or LVFX at 37 °C for 24 h. The control wells without antimicrobial agents contained an equivalent amount of each solvent. The CFUs of biofilms were calculated as described above.

### 2.7. Treatment of Candida Culture Supernatant

We investigated whether materials in the *Candida* culture supernatant (with inactivated antibacterial drugs) were heat-sensitive or could be inactivated by a hydrolytic enzyme. For the heat treatment, supernatants were heated at 100 °C for 30 min. Hydrolytic enzyme treatment was performed by activating the supernatants with 50 μg/mL lyticase (β1,3-_D_-glucanase) at 25 °C for 8 h, 50 μg/mL DNase at 37 °C for 8 h, and 50 μg/mL proteinase K at 37 °C for 8 h (all materials from Merck KGaA, Darmstadt, Germany) and then inactivated at 100 °C for 10 min. In addition, we performed the drug efflux pump inhibitor treatment by adding phenylalanine-arginine-β-naphthylamide (PAβN, Merck KGaA) to the supernatant to determine whether the materials were excreted by *E*. *coli*. The *E*. *coli* biofilms were formed in the presence of 50 μg/mL MEPM and PAβN-treated supernatants (12.5, 25, 50, and 100% *v*/*v*) at 37 °C for 24 h, and the CFUs were counted as described above. The wells without MEPM were used as control and contained an equivalent amount of DMSO.

### 2.8. Fractionation of Candida Supernatant

The culture supernatant of the *Candida* biofilm was centrifugally fractionated into solutions of <10 kDa and >10 kDa. Briefly, *Candida* supernatants were put into an ultrafiltration column (Amicon Ultra-4 10K, Merck KGaA) and centrifuged at 6000× *g* for 20 min. The flow-through solution was used as <10 kDa fraction, and the concentrate fraction inside the column was used as >10 kDa fraction. The >10 kDa fraction was diluted with RPMI 1640 supplemented with MOPS medium before use. *E*. *coli* biofilms were formed in the presence of each supernatant fraction and 50 μg/mL MEPM at 37 °C for 24 h. The control wells without supernatant fractions contained an equivalent amount of DMSO. The *E*. *coli* CFUs were counted as described above.

### 2.9. Statistical Analysis

The significance of differences between groups in the dose–response experiments was calculated using the Kruskal–Wallis test. The significance of differences between groups in all other tests was analyzed by Tukey’s test using one-way ANOVA. A *p*-value of <0.05 was considered significant. All statistical analyses were performed using Prism 9.1.2 (GraphPad Software, LLC, San Diego, CA, USA, https://www. graph pad.com/scientificsoftware/prism/ (accessed on 1 October 2022)).

## 3. Results

### 3.1. Biofilm Formation by E. coli Isolates

Of the 52 isolates, 14 (26.9%) formed biofilms: four strains exhibited high and 10 strains exhibited low biofilm-forming capacities (Figure 1). All four strains exhibiting high biofilm-forming capacity were from patients with UTIs, while the other 10 strains with low biofilm-forming capacity were from patients with various conditions.

### 3.2. Drug Susceptibilities of E. coli Planktonic Cells and Biofilms

Antimicrobial susceptibility testing was performed using MEPM, CMZ, CTRX and LVFX against *E. coli*, the drugs used to treat bacteremia caused by *E. coli*. The planktonic cells of all 52 strains were sensitive to CMZ and MEPM. However, 14 (26.9%) and 11 (21.2%) strains were resistant to CTRX and LVFX, respectively (Figure 1). Of these 25 strains, nine were resistant to both CTRX and LVFX. From the four strains exhibiting high biofilm-forming capacity, RB-3 formed the most stable biofilms and was used as a representative for the drug susceptibility testing of biofilms. In the presence of antimicrobial drugs, the viable cell counts of RB-3 decreased in a dose-dependent manner compared to the controls. The maximum reduction was observed at a drug concentration of 50 μg/mL, and the CFU reached 10^−3^–10^−4^/mL (Figure 2).

### 3.3. Drug Susceptibilities of Dual-Species Biofilms with E. coli and C. albicans

Since the four drugs showed similar behavior against an *E. coli* single biofilm, we tested the effect of MEPM, which can also be used against ESBL-producing bacteria and has high clinical significance, in a dual-species biofilm with *C. albicans*. The MEPM susceptibilities of the dual-species biofilm (*E*. *coli* RB-3 and *C*. *albicans*) and single-species biofilm (*E*. *coli* RB-3 or *C*. *albicans* only) were calculated using the CFU assay (Figure 3). For the single-species biofilms containing *E*. *coli*, the cell count decreased by 10^−^^4^-fold in the presence of 50 μg/mL MEPM compared with the control (no MEPM) (*p* < 0.05). In contrast, the *E*. *coli* CFU count in the dual-species biofilm in the presence of MPEM was 1000-fold lower than that in the control in the absence of MPEM (*p* < 0.05). However, MEPM did not affect the CFU count of *C*. *albicans,* either in dual-species or single-species biofilms.

### 3.4. Effect of C. albicans Culture Supernatant on E. coli Biofilm Formation in the Presence of Antimicrobial Agents

We observed that MEPM only slightly decreased the viable cell count of *E*. *coli* RB-3 in dual-species biofilms. Therefore, we considered the *Candida* culture supernatant possibly contained substances that impaired the antimicrobial effects of MEPM. Therefore, we mixed antimicrobial agents (MEPM, CMZ, CTRX, and LVFX) with the *Candida* supernatant. In a supernatant-free medium, the viable *E*. *coli* count decreased when any of the four agents were added; however, all drugs were tolerated in the presence of *Candida* supernatant (Figure 4). In the presence of LVFX, the antimicrobial tolerance-inducing effect of the culture supernatant was lower than that in the presence of the other three drugs.

### 3.5. Effects of Various Treatments on the Candida Culture Supernatant

We demonstrated that the *Candida* supernatant reduced the effects of antibacterial drugs on *E*. *coli* RB-3 biofilms in a concentration-dependent manner and did not change after heating at 100 °C for 30 min (Figure 5a). Furthermore, treatment with lyticase (which degrades yeast cell wall glucan) or DNase and proteinase K did not decrease the inhibitory effects of the *Candida* supernatant in dual-species biofilms treated with MEPM (Figure 5b). We also added the PAβN pump inhibitor to the culture supernatant to explore whether *E*. *coli* effluxed the drugs; however, the inhibitory effects of the *Candida* supernatant were not altered (Figure 6).

### 3.6. Effect of Candida Culture Supernatant Fractions on E. coli Biofilm Formation

The two supernatant fractions (<10 kDa and >10 kDa) of *Candida* culture supernatant were tested individually. The <10 kDa fraction + MEPM did not exhibit an antibacterial effect against *E*. *coli* RB-3 biofilms, whereas the >10 kDa fraction + MEPM showed an antibacterial effect (Figure 7). Therefore, we inferred that the molecular weight of the substance that induces MEPM tolerance in *E. coli* biofilms was <10 kDa in size.

## 4. Discussion

*E. coli* is the major causative agent of UTIs, and its isolation from patients with BSIs has also been steadily increasing. In this study, we focused on the biofilm-forming capacity of these isolates. Of the 52 isolates examined, 14 (26.9%) formed biofilms, and four of the those (28.6%) demonstrated a high biofilm-forming capacity. Although all isolates examined in this study were from a single hospital, this rate was consistent with the rates reported in other studies [16]. We demonstrated that the planktonic cells of the RB-3 strain of *E. coli*, with a high biofilm-forming ability, were sensitive to MEPM, CMZ, CTRX, and LVFX in an in vitro drug susceptibility test. However, in a drug susceptibility test under the same conditions, strain RB-3 readily formed biofilms, and although the number of viable cells decreased in a dose-dependent manner, approximately 10^5^ cells survived. These results suggest that RB-3 resists the effects of antibacterial drugs by forming biofilms.

Previous studies have reported that biofilms containing both *E. coli* and *C. albicans* increase the hyphal formation and virulence [26]. Our results indicated that the antibacterial drug tolerance of *E. coli* was induced in dual-species biofilms. The protective effect of *C. albicans* in dual-species biofilms has been reported with various counterpart bacteria [24,27,28,29]. For example, *Streptococcus aureus*, *S. mutans,* and *P. aeruginosa* increased tolerance against vancomycin, chlorhexidine, and MEPM, respectively. Although *E. coli* is reportedly tolerant to ofloxacin, a quinolone drug in dual-species biofilms containing *C. albicans* [24], this is the first report investigating the tolerance of *E. coli* to β-lactams, such as MEPM. We demonstrated that the culture supernatant of *C. albicans* induced an antimicrobial tolerance to the cell wall synthesis inhibitor MEPM, but only partial tolerance to LVFX, a quinolone. By identifying the tolerance-inducing factor in the culture supernatant of *C. albicans* and evaluating the mechanism responsible, it would be possible to explain the differences between its varying effects against MEPM and LVFX.

Biofilms feature a three-dimensional architecture of microbial aggregates embedded in an extracellular matrix (ECM), which is composed of polysaccharides, proteins, nucleic acids, and lipids [30,31]. *Candida* hyphal formation endows a biofilm with high-order architecture that may inhibit antimicrobial entry into cells. The ECM produced by *C. albicans* consists of β-_D_-glucan, proteins, nucleic acids, and lipids [32]. The culture supernatant isolated from the *Candida* biofilm displayed tolerance to the drugs; therefore, it can be inferred that the responsible material might have been secreted by *Candida* cells. Studies have proven that *Candida* extracellular matrix components (β-_D_-glucan or mannan) can induce tolerance to antimicrobial drugs in dual-species biofilms [20,24,27]. Lopez-Medina et al. demonstrated how proteins released by *C. albicans* suppress the expression of the iron uptake machinery and reduce the virulence of the Gram-negative bacterium *Pseudomonas aeruginosa* [21]. Our study showed increased tolerance not only to MEPM, but also to other lactam and quinolones, although the tolerance to different drugs varied to some extent. In contrast, Alam et al. [27] reported that *P. aeruginosa* increased tolerance specifically against MEPM (not seen in other β-lactam), whereas de Brucker K et al. [24] reported that *E. coli* increased tolerance specifically to ofloxacin (not seen in kanamycin). Taken together, these findings suggest that the tolerance mechanism in the present study is different from the previous studies. Furthermore, our study demonstrated that treatment of the culture supernatant with hydrolytic enzymes (lyticase, DNase, and proteinase K) did not restore antimicrobial activity. This result suggests that the active components released by *C. albicans* are not lyticase-degraded β-_D_-glucan, DNase-degraded extracellular DNA, or protease-degraded proteins. Furthermore, the induction effect of antibacterial drug tolerance did not reduce despite the supernatant heat treatment and the addition of an *E*. *coli* efflux pump inhibitor. Nevertheless, the centrifugal fractionation of the *Candida* culture supernatants revealed that the molecular weight of the active component was <10 kDa. Therefore, further studies are needed to explore whether the tolerance-inducing effect is *E. coli*-specific.

The drugs (MEPM, CMZ, CTRX, and LVFX) used to treat *E*. *coli* BSIs include carbapenems, cephamycins, cephalosporins, and fluoroquinolones, which act via diverse mechanisms. As the unknown <10 kDa compound induces tolerance to all of these agents, the mechanism responsible should be explored. As the compound was present in the culture supernatants of both *Candida* biofilms and planktonic culture (Appendix A), the material was not biofilm-specific and, therefore, might also affect the antimicrobial treatment of infections that are not associated with biofilms. Researchers have shown that farnesol, a small molecule secreted by *C. albicans,* alters the expression of several genes involved in cell membrane formation and lipid synthesis, cell morphological features, and impairs the biofilm formation capacities in the Gram-negative bacterium *Acinetobacter baumannii* [22] and Gram-positive bacterium *Staphylococcus aureus* [28]. Therefore, future studies to purify and characterize the active component (<10 kDa) in *Candida* culture supernatant using electron microscopy are essential to gain further insights. Furthermore, a previous study has shown that *C. albicans* alters gene expression in *P. aeruginosa* and affects its iron uptake mechanism [21]. As *E. coli* also possess an iron-dependent biofilm formation mechanism [33], we assumed that the active compound may have altered gene expression in *E. coli* by altering the iron-dependent biofilm formation mechanism. Revealing the molecular mechanisms of action of the active compound produced by *C. albicans* on drug resistance of *E. coli* will be an important subject.

Biofilm infections are generally difficult to treat; therefore, careful drug selection is important. For example, in biofilm infections with *S*. *aureus*, biofilm formation is enhanced by sub-therapeutic doses of β-lactams [34]. Ruiz-Sorribas et al. [35] explored the effects of antimicrobial agents on single-, dual-, and three-species biofilms of *S*. *aureus*, *E*. *coli*, and *C*. *albicans*. It was discovered that the agents were not very effective when used alone; however, when combined, they exerted synergistic effects. Therefore, these studies have emphasized the importance of careful drug selection based on the microbes to be controlled. Our study implicated that the standard treatment may become insufficient for *E. coli* bacteremia coinfection with *C. albicans*. In such cases, the combination of antibacterial and antifungal agents effective against *C. albicans* biofilms may be a better option. Although this study revealed that dual-species biofilm of *C. albicans* and *E. coli* leads to antimicrobial resistance in *E. coli*, the effect of *C. albicans* on antifungal drug resistance and the therapeutic effect of the combination of antimicrobials and antifungal drugs have not been verified. The validation in animal studies of whether the combination of antibacterial and antifungal agents is effective in treating infections caused by dual-species biofilms is an important issue for further investigation.

In the present study, we did not identify the substance that induced tolerance to antimicrobials or the underlying mechanism. However, the *Candida* culture supernatant-induced tolerance to antibacterial agents will serve as a foundation for further studies on treatments for multi-species biofilm infections.

## 5. Conclusions

In conclusion, we demonstrated that the *Candida* culture supernatant induced antibacterial tolerance in *E. coli* when present in dual-species biofilms with *C. albicans*; however, we did not identify the exact factor responsible for tolerance induction in this study. We believe that this research may facilitate the development of new treatments for polymicrobial biofilm infections.

## Figures and Tables

**Figure 1 microorganisms-10-02179-f001:**
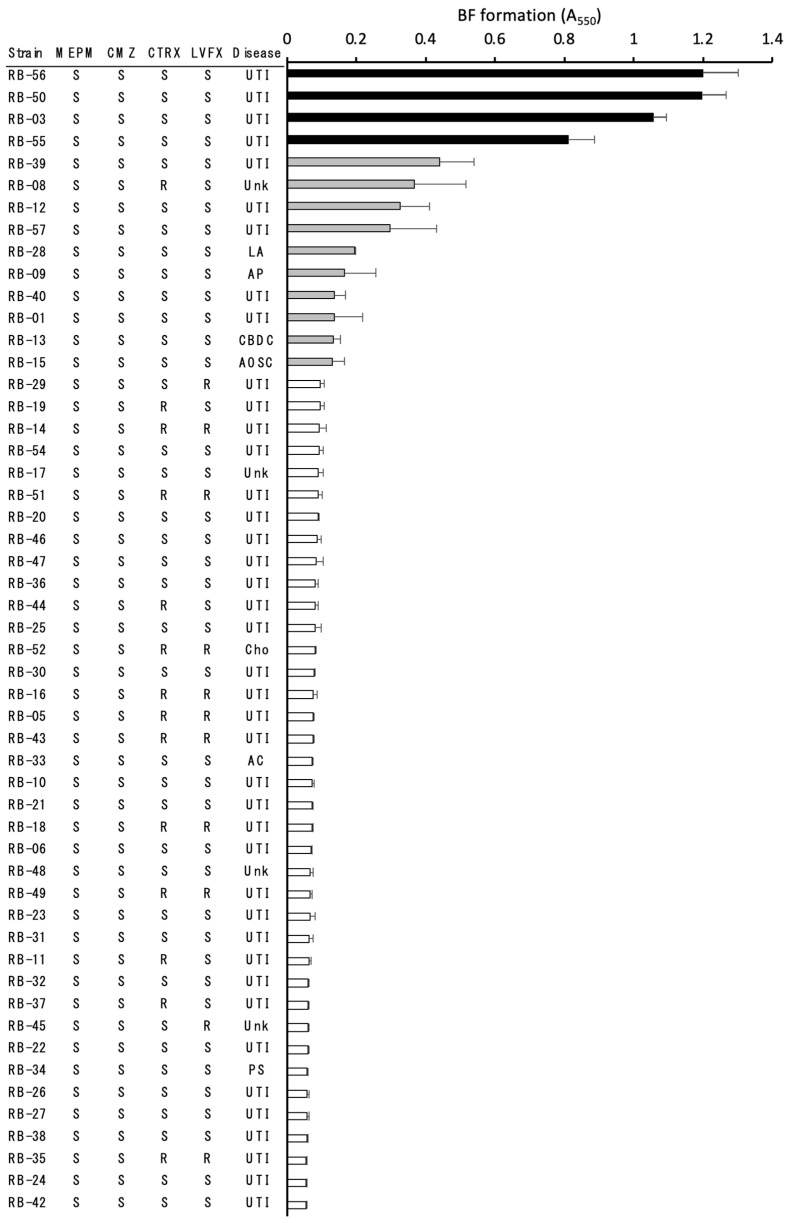
Biofilm-forming capacities and drug susceptibilities of *E. coli* clinical isolates. Black, gray, and white bars indicate high-capacity (A_550_ > 0.8), low-capacity (A_550_ 0.1–0.8), and non-biofilm-forming strains (A_550_ < 0.1), respectively. Measurements were performed three times, and the data shown are the means ± standard deviations. Antimicrobial drug susceptibilities from left to right: meropenem (MEPM), cefmetazole (CMZ), ceftriaxone (CTRX), and levofloxacin (LVFX) of planktonic *E. coli* cells; the primary diseases are shown next to the strain numbers. AC, acute cholangitis; AOSC, acute obstructive suppurative cholangitis; AP, acute prostatitis; CBDC, common bile duct calculi; Cho, cholecystitis; LA, liver abscess; PS, pyogenic spondylitis; Unk, Unknown; UTI, urinary tract infection. S, susceptible; R, resistant. The drug susceptibilities of planktonic *E*. *coli* cells were performed following the Clinical and Laboratory Standards Institute (CLSI) M100-Ed31 protocol. A_550_ of negative controls without antimicrobial agents was 0.06 (range: 0.055–0.087).

**Figure 2 microorganisms-10-02179-f002:**
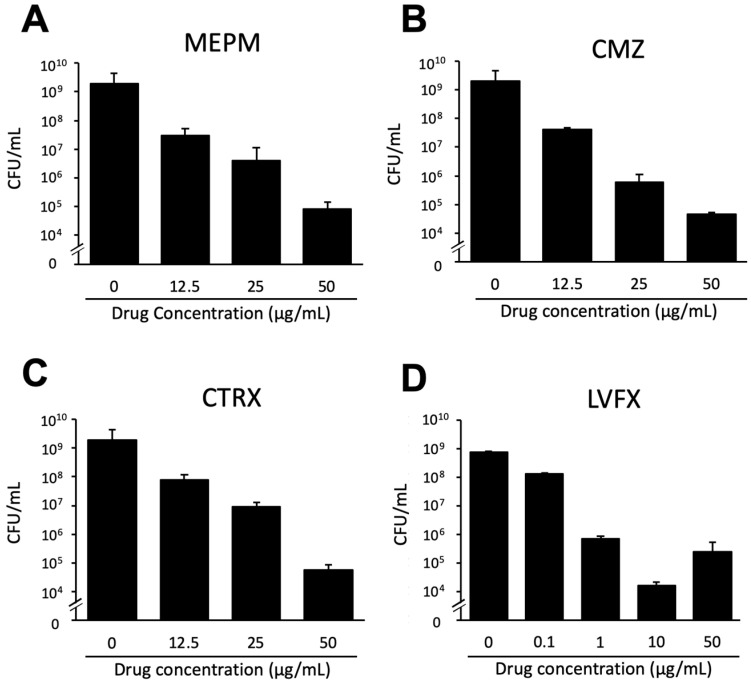
Drug susceptibilities of an *E. coli* RB-3 biofilm to four antibacterial agents. A high-capacity biofilm-forming strain, RB-3, was grown under biofilm-forming conditions for 24 h. Various concentrations of MEPM, CMZ, CTRX (0, 12.5, 25, and 50 μg/mL), and LVFX (0, 0.1, 1.0, 10, and 50 μg/mL) were added followed by incubation at 37 °C for 24 h (**A**–**D**). The wells without drugs (0 μg/mL) were used as controls. Then, the viable cells were counted. The data are the means ± standard deviations of three measurements. MEPM, meropenem; CMZ, cefmetazole; CTRX, ceftriaxone; LVFX, levofloxacin. Statistically significant differences between groups were evaluated using the Kruskal–Wallis test; *p* < 0.05. The *p*-value of each figure was less than 0.05.

**Figure 3 microorganisms-10-02179-f003:**
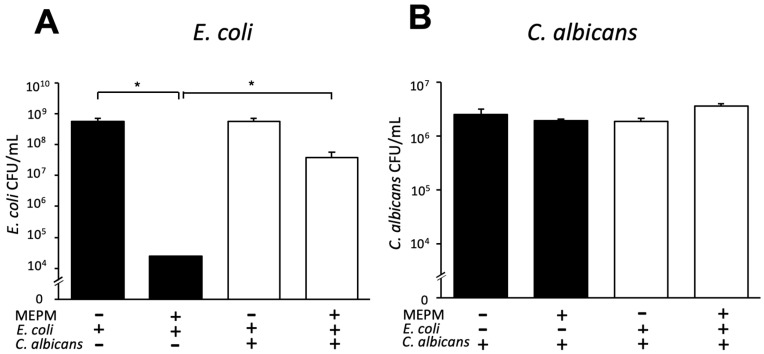
Antimicrobial effects of MEPM against a dual-species biofilm with *E. coli* and *C. albicans*. (**A**) Survival of *E. coli* RB-3 in single (black) and dual-species (white) biofilms treated with or without MEPM (50 µg/mL); only *E. coli* was quantified using a selective medium containing micafungin. (**B**) Survival of *C. albicans* in single (black) and dual-species (white) biofilms treated with or without MEPM (50 µg/mL); only *C. albicans* was quantified using a selective medium containing streptomycin. Measurements were performed three times, and the data are the means ± standard deviations. *n* = 3/group. Statistically significant differences between groups were evaluated using Tukey’s test with one-way ANOVA; * *p* < 0.05.

**Figure 4 microorganisms-10-02179-f004:**
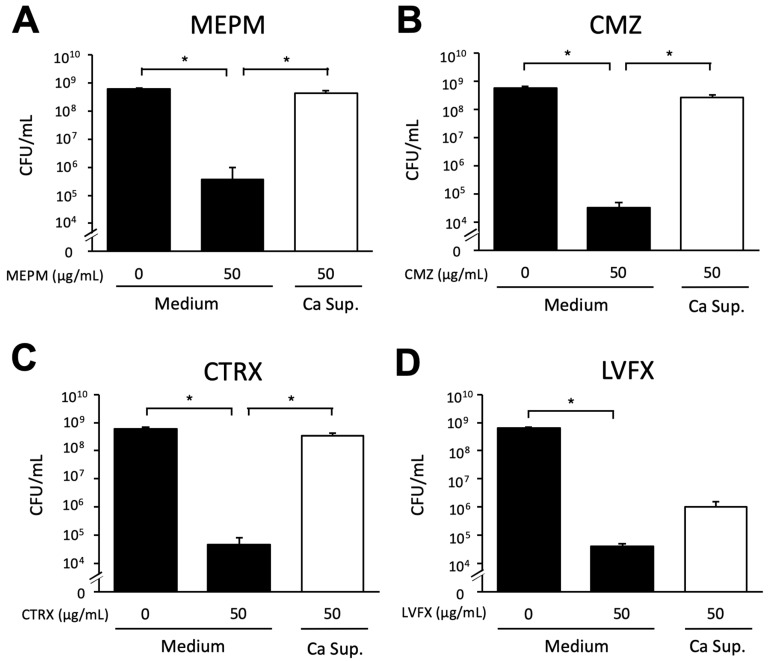
Effect of *C*. *albicans* culture supernatant on *E*. *coli* biofilm formation in the presence of antimicrobial agents. *E*. *coli* RB-3 survival in the presence of *Candida* culture supernatant (Ca Sup.) and 0 or 50 μg/mL MEPM, CMZ, CTRX, or LVFX (**A**–**D**). Measurements were performed three times, and the data are presented as the mean ± standard deviation. *n* = 3/group. Statistically significant differences between groups were evaluated using Tukey’s test with one-way ANOVA; * *p* < 0.05.

**Figure 5 microorganisms-10-02179-f005:**
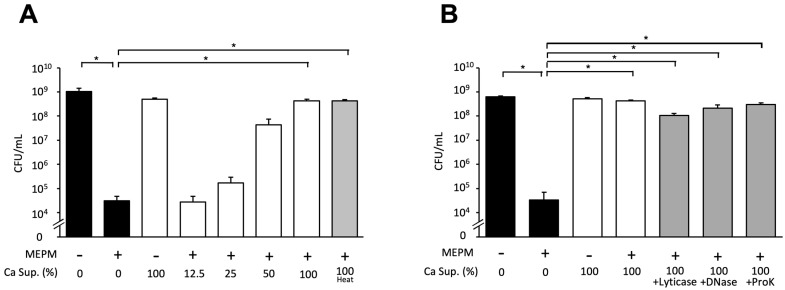
Effects of *Candida* culture supernatant treated with heat and hydrolytic enzymes on *E. coli* biofilm formation in the presence of MEPM. (**A**) *E. coli* RB-3 survival in the presence of 50 μg/mL MEPM and various concentrations of *Candida* culture supernatant (Ca Sup.; 0%, 12.5%, 25%, 50%, and 100% *v*/*v*) under each treatment and heating at 100 °C for 30 min; (**B**) culture supernatant (Ca Sup.) was treated with 50 μg/mL lyticase (β1,3-_D_-glucanase) at 25 °C for 8 h, 50 μg/mL DNase at 37 °C for 8 h, or 50 μg/mL proteinase K at 37 °C for 8 h. Measurements were performed three times and the data are the means ± standard deviations. *n* = 3/group. Statistically significant differences between groups were evaluated using Tukey’s test with one-way ANOVA; * *p* < 0.05.

**Figure 6 microorganisms-10-02179-f006:**
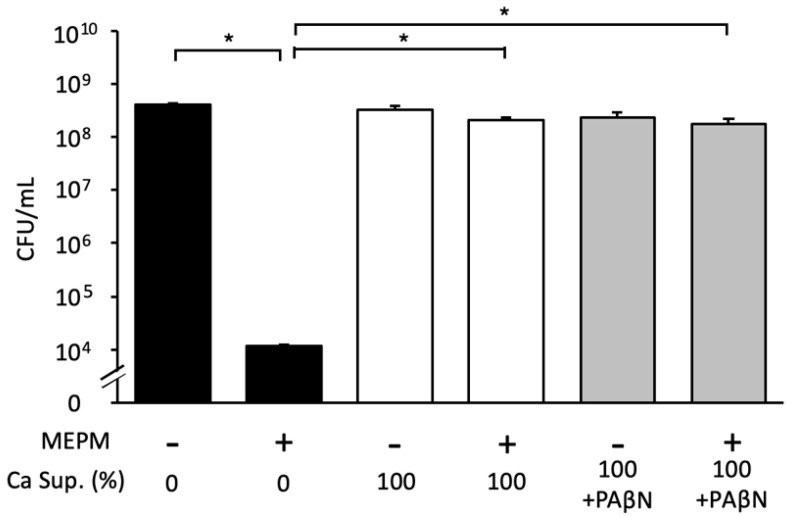
Effect of *Candida* culture supernatant (Ca Sup.) treated with drug efflux pump inhibitors on *E*. *coli* biofilm formation in the presence of MEPM. The drug efflux pump inhibitor PAβN was added to the supernatant. Measurements were performed three times, and the data are presented as the mean ± standard deviation. *n* = 3/group. Statistically significant differences between groups were evaluated using Tukey’s test with one-way ANOVA; * *p* < 0.05.

**Figure 7 microorganisms-10-02179-f007:**
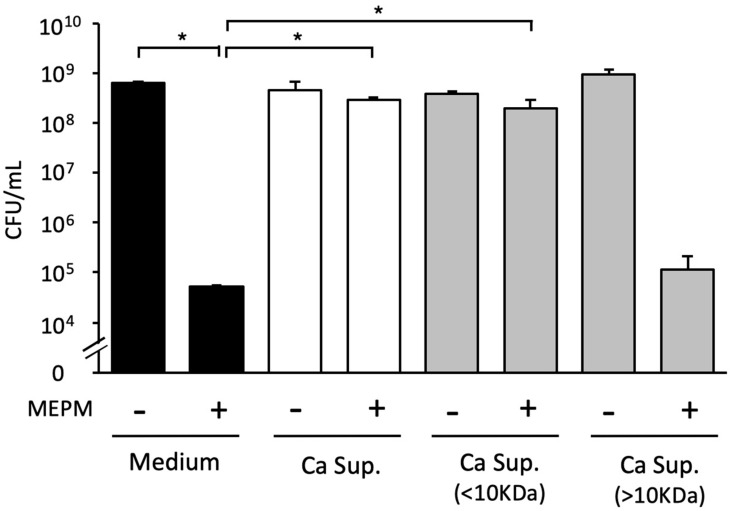
Effects of *Candida* culture supernatant (Ca Sup.) fractions on *E. coli* biofilm formation. The effects of supernatant fractions containing materials <10 kDa and >10 kDa in size in the presence of 50 μg/mL MEPM on *E. coli* survival. The measurements were performed three times, and the data are presented as the mean ± standard deviation. *n* = 3/group. Statistically significant differences between groups were evaluated using Tukey’s test with one-way ANOVA; * *p* < 0.05.

## Data Availability

Essential data supporting the reported results were included in this study. All other Appendix A are available upon request from the corresponding author.

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
