# Peer review of "Candida albicans Promotes the Antimicrobial Tolerance of Escherichia coli in a Cross-Kingdom Dual-Species Biofilm"

_microorganisms, 2022, doi:10.3390/microorganisms10112179_

Round 1
Reviewer 1 Report
Authors discovered that some components of C. albicans biofilms increase tolerance of E. coli to antimicrobials. As authors mentioned it was previously described that components of EPS may have such effects, however present work points at some other factors (<10kDa). The reseearch is designed very thoroughly. Methods are described properly. I would like to know more about centrifugal fractionation - what the material preparation looks like, what is the procedure?
Results are presented very clearly and support the conclusions. Overall, I found the manuscript well written and scientifically interesting
Author Response
#Reviewer 1
Question 1: Authors discovered that some components of C. albicans biofilms increase tolerance of E. coli to antimicrobials. As authors mentioned it was previously described that components of EPS may have such effects, however present work points at some other factors (<10kDa). The research is designed very thoroughly. Methods are described properly. I would like to know more about centrifugal fractionation - what the material preparation looks like, what is the procedure?
Results are presented very clearly and support the conclusions. Overall, I found the manuscript well written and scientifically interesting
Response: Dear Reviewer, thank you for your time and efforts in evaluating our manuscript. We are grateful for your positive feedback. We also appreciate your insightful comment on centrifugal fractionation. We have updated the method of fractionation of active compounds in the revised manuscript, as shown below. Please see Lines 159-163.
The culture supernatant of the Candida biofilm was centrifugally fractionated into solutions of <10 kDa and >10 kDa. Briefly, Candida supernatants were put into an ultrafiltration column (Amicon Ultra-4 10K, Merck KGaA) and centrifuged at 6,000 ×g for 20 min. The flow-through solution was used as <10 kDa fraction, and the concentrate fraction inside the column was used as >10 kDa fraction. The >10 kDa fraction was diluted with RPMI 1640 supplemented with MOPS medium before use.

Reviewer 2 Report
1- The authors have to photograph the formed biofilms by Scanning Electron Microscope, I think one phenotypic analysis to assume high or low biofilm forming capacity is not enough
2- The author have to add more controls to compare biofilm formation rather than E. coli (P. aeruginosa or Proteus) to confirm or exclude the dual-species biofilms
3- The authors have to discuss the dual species biofilms in the context of other bacterial species based on the proposed controls (to answer whether this Candida derived tolerance inducing factor is specific to E. coli or general to Enterobacteriaceae)
4- The authors have to discuss properly the nature of this tolerance inducing factor based on their biochemical approaches (Proteinase K, DNase,......etc)
5- I recommend citing these references in the introduction and the methods section
A- Abdel-Halim, M. S., Askoura, M., Mansour, B., Yahya, G., & El-Ganiny, A. M. (2022). In vitro activity of celastrol in combination with thymol against carbapenem-resistant Klebsiella pneumoniae isolates. The Journal of antibiotics, 10.1038/s41429-022-00566-y
B- Yahya, G., Ebada, A., Khalaf, E. M., Mansour, B., Nouh, N. A., Mosbah, R. A., Saber, S., Moustafa, M., Negm, S., El-Sokkary, M., & El-Baz, A. M. (2021). Soil-Associated Bacillus Species: A Reservoir of Bioactive Compounds with Potential Therapeutic Activity against Human Pathogens. Microorganisms, 9(6), 1131. https://doi.org/10.3390/microorganisms9061131
C- Yahya, G., Ebada, A., Khalaf, E. M., Mansour, B., Nouh, N. A., Mosbah, R. A., Saber, S., Moustafa, M., Negm, S., El-Sokkary, M., & El-Baz, A. M. (2021). Soil-Associated Bacillus Species: A Reservoir of Bioactive Compounds with Potential Therapeutic Activity against Human Pathogens. Microorganisms, 9(6), 1131. https://doi.org/10.3390/microorganisms9061131
Author Response
# Reviewer 2
Question 1: The authors have to photograph the formed biofilms by Scanning Electron Microscope, I think one phenotypic analysis to assume high or low biofilm forming capacity is not enough.
Response: Dear Reviewer, thank you for your time and efforts in evaluating our manuscript. We are grateful for your insightful suggestions. We agree with your opinions and have revised the manuscript accordingly. We believe the incorporations have further strengthened our manuscript.
As you pointed out, we feel that an electron microscope observation is important for understanding the drug resistance mechanism of E. coli in the presence of C. albicans. In the future, when the active ingredient is identified, we plan to observe it with an electron microscopic analysis to evaluate the effect of that compound.
We have described this in the discussion section as follows:
Therefore, future studies to purify and characterize the active component (<10 kDa) in Candida culture supernatant using electron microscopy are essential to gain further insights. (Lines 349-351)
Question 2: The authors have to add more controls to compare biofilm formation rather than E. coli (P. aeruginosa or Proteus) to confirm or exclude the dual-species biofilms
Response: Thank you for this insightful suggestion. We have added the following sentences according to your suggestion:
The protective effect of C. albicans in dual-species biofilms has been reported with various counterpart bacteria [24, 27–29]. For example, Streptococcus aureus, S. mutans, and P. aeruginosa increased tolerance against vancomycin, chlorhexidine, and MEPM, respectively. (Lines 301-304)
Accordingly, we have added Reference # 29 to the text and updated the list.
Question 3: The authors have to discuss the dual species biofilms in the context of other bacterial species based on the proposed controls (to answer whether this Candida derived tolerance inducing factor is specific to E. coli or general to Enterobacteriaceae)
Response: Thank you for this suggestion. We have added the following sentences to incorporate your suggestions:
Our study showed increased tolerance not only to MEPM but also to other -lactam and quinolones, although the tolerance to different drugs varied to some extent. In contrast, Alam et al. [27] reported that P. aeruginosa increased tolerance specifically against MEPM (not seen in other β-lactam), whereas de Brucker K et al. [24] reported E. coli increased tolerance specifically to ofloxacin (not seen in kanamycin). Taken together, these findings suggest that the tolerance mechanism in the present study is different from the previous studies. Furthermore, our study demonstrated that treatment of the culture supernatant with hydrolytic enzymes (lyticase, DNase, and proteinase K) did not restore antimicrobial activity. This result suggests that the active components released by C. albicans are not lyticase-degraded β-D-glucan, DNase-degraded extracellular DNA, or protease-degraded proteins.Furthermore, the induction effect of antibacterial drug tolerance did not reduce despite supernatant heat treatment and the addition of an E. coli efflux pump inhibitor. Nevertheless, the centrifugal fractionation of the Candida culture supernatants revealed that the molecular weight of the active component was <10 kDa. Therefore, further studies are needed to explore whether the tolerance-inducing effect is E. coli-specific.
(Lines 323-329, 331-333, 337-338)
Question 4: The authors have to discuss properly the nature of this tolerance inducing factor based on their biochemical approaches (Proteinase K, DNase,......etc)
Response: Thank you for highlighting this. Accordingly, we have added the following sentence to the discussion section:
This result suggests that the active components released by C. albicans are not lyticase-degraded β-D-glucan, DNase-degraded extracellular DNA, or protease-degraded proteins.
(Lines 331-333)
Question 5: I recommend citing these references in the introduction and the methods section
A- Abdel-Halim, M. S., Askoura, M., Mansour, B., Yahya, G., & El-Ganiny, A. M. (2022). In vitro activity of celastrol in combination with thymol against carbapenem-resistant Klebsiella pneumoniae isolates. The Journal of antibiotics, 10.1038/s41429-022-00566-y
B- Yahya, G., Ebada, A., Khalaf, E. M., Mansour, B., Nouh, N. A., Mosbah, R. A., Saber, S., Moustafa, M., Negm, S., El-Sokkary, M., & El-Baz, A. M. (2021). Soil-Associated Bacillus Species: A Reservoir of Bioactive Compounds with Potential Therapeutic Activity against Human Pathogens. Microorganisms, 9(6), 1131. https://doi.org/10.3390/microorganisms9061131
C- Yahya, G., Ebada, A., Khalaf, E. M., Mansour, B., Nouh, N. A., Mosbah, R. A., Saber, S., Moustafa, M., Negm, S., El-Sokkary, M., & El-Baz, A. M. (2021). Soil-Associated Bacillus Species: A Reservoir of Bioactive Compounds with Potential Therapeutic Activity against Human Pathogens. Microorganisms, 9(6), 1131. https://doi.org/10.3390/microorganisms9061131
Response: Thank you for providing us with these important studies. We have added the suggested studies. However, your recommended references, B and C are the same; therefore, we have added two references (18 for A and 19 for B) to the Introduction section. Please see below.
Identifying compounds that are effective against drug-resistant bacteria and compounds that reduce drug resistance [18, 19] and research related to drug resistance caused by microbial components [20-22] were reported.
(Lines 55-57)
Reviewer 3 Report
microorganisms-1965196
Candida albicans promotes the antimicrobial tolerance of Escherichia coli in a cross-kingdom dual-species biofilm
Authors described the impact on C. albicans to antimicrobial tolerance of E. coli. I have a few doubts and questions to authors about the manuscript
Firstly, i have not seen results from controls. In some case you put results from untreated strain, but how about the solvent control?
Next, lack of statistic analysis in some results. Please put some detail in proper places. What is more important, I do not think that t-Student test is proper for your analysis, I strongly recommend do use ANOVA to compare results.
And what is the most important, the research are not clearly presented. I have some doubts why did you choose this specific strain to research, also sometimes you use one drug and other time fours, without comment why you did it. Please clarify it.
Minor objection:
L51- Please put the hypothesis.
L59 - Please add more details about the methodology of identification.
L88 - What was the concentration of antimicrobial substances? Also put some detail about the non-treated and solvent control
L114 - Please put some information about the non-treated and solvent control
Figure 1 - Please put information:
- data from controls;
- statistical analysis;
- how did you classify the susceptibility? and what concentrations of drugs did you use?
L156 - Please clarify the sentence. I do not understand why did you choose the RB-3 strain? In my opinion RB-56 or RB-50 have better BF parameter and they are sensitive do drug treatment also. However, without statistical analysis and clear information how did you classify the the susceptibility/resistanace is difficult to compare.
Figure 2 - Please put the information about control and statistical analysis. Also why did you use different concentration of LVFX?
Figure 3 - Why did you show results only for MEPM treatment?
Figure 4 - Please change the names: "Medium" and "supernatant" because it is confusing, if I understand properly you treated the E. coli medium with fungal supernatant? But now it looks as you treated the medium and supernatant of E. coli
Author Response
#Reviewer 3
Question 1: Authors described the impact on C. albicans to antimicrobial tolerance of E. coli. I have a few doubts and questions to authors about the manuscript
Firstly, i have not seen results from controls. In some case you put results from untreated strain, but how about the solvent control?
Response: Dear Reviewer, thank you for your time and efforts in evaluating our manuscript. We are grateful for your insightful suggestions. We agree with your opinions and have revised the manuscript accordingly. We believe the suggestions have further strengthened our manuscript.
Regarding the controls, the same amount of solvent in which the drug was dissolved was added to the wells in respective assessments and used drug-free controls. We apologize for missing this information in our original manuscript. Nevertheless, we have updated the text mentioning the controls, as appropriate.
The wells with 0 µg/mL drug contained an equivalent amount of each solvent and were used as drug-free controls. (Lines 103-104)
The control wells contained an equivalent amount of DMSO. (Lines 115-116)
The control wells without supernatant fractions contained an equivalent amount of DMSO. (Lines 164-165)
Question 2: Next, lack of statistic analysis in some results. Please put some detail in proper places. What is more important, I do not think that t-Student test is proper for your analysis, I strongly recommend do use ANOVA to compare results.
Response: Thank you for your input. We agree with your suggestion and have performed ANOVA to compare the statistical differences. We have revised the “Statistical analysis” section in the “Materials and Methods” and mentioned the appropriate statistical analyses performed for each data set in respective “Figure legends”.
2.9. Statistical analysis
The significance of differences between groups in the dose-response experiments was calculated using the Kruskal–Wallis test. The significance of differences between groups in all other tests was analyzed by Tukey’s test using one-way ANOVA. A P-value of < 0.05 was considered significant. All statistical analyses were performed using Prism 9.1.2 (GraphPad Software, LLC, San Diego, CA, USA, https:// www. graph pad. com/scientific -software/prism/). (Lines 168-173)
Figure legends
Figure 2
Statistically significant differences between groups were evaluated using the Kruskal–Wallis test; P < 0.05. The P-value of each figure was less than 0.05. (Lines 212-214)
Figures 3, 4, 5, 6, 7, and 7
Statistically significant differences between groups were evaluated using Tukey’s test with one-way ANOVA; *P < 0.05.
(Lines 233-234, 249-250, 267-268, 273-274, 285-286)
Question 3: And what is the most important, the research is not clearly presented. I have some doubts why did you choose this specific strain to research, also sometimes you use one drug and other time fours, without comment why you did it. Please clarify it.
Response: We are extremely sorry for not being clear in representing the contents. At the same time, we are thankful to you for bringing up this important question. Of the 52 E. coli strains used, 4 showed high-biofilm-forming capacity. Of these four strains, we selected RB-3 for further analyses as it formed the most stable biofilm. In addition, the drugs used in this study are used to treat bacteremia caused by E. coli. Among them, a carbapenem, MEPM, is used when an infection caused by ESBL-producing bacteria is suspected. In recent years, with the increase in cases of infection with ESBL-producing bacteria, it is speculated that there will be more opportunities to administer MEPM empirically. Therefore, in this study, we focused on MEPM in particular.
To ensure clarity, we have revised the contents appropriately. Please see the changes below and in the manuscript.
From the four strains exhibiting high biofilm-forming capacity, RB-3 formed the most stable biofilms and was used as a representative for the drug susceptibility testing of biofilms. (Line 201)
Since the four drugs showed similar behavior against E. coli single biofilm, we tested the effect of MEPM, which can also be used against ESBL-producing bacteria and has high clinical significance, in a dual-species biofilm with C. albicans. (Lines 216-218)
Minor objection:
Question 4: L51- Please put the hypothesis.
Response: Thank you for highlighting this. We have revised the Introduction section to put our hypothesis.
Since we hypothesized that E. coli clinically isolated from a Japanese hospital might contain biofilm-forming strains, the biofilm-forming capacity of E. coli isolates from the blood samples of patients was tested. Identifying compounds that are effective against drug-resistant bacteria and compounds that reduce drug resistance [18, 19] and research related to drug resistance caused by microbial components [20-22] were reported. We also hypothesized that C. albicans might produce substances that affect drug resistance in biofilm-forming E. coli. (Lines 53-59)
Question 5: L59 - Please add more details about the methodology of identification.
Response: We have added the sentences according to your suggestion.
The isolates were initially identified using a MicroScan system (Auto SCAN4; Beckman Coulter, Brea, CA) in conjunction with Neg combo panels. Identification was confirmed via 16S rRNA gene sequencing [23]. The microbial 16S rRNA gene was amplified using PCR with primers 27F (5’-GAGTTTGATCCTGGCTCAG-3’) and 1492R (5’-GGYTACCTTGTTACGACTT-3’) and was sequenced. Strains with >99% sequence similarity were defined as conspecific. (Lines 66-71)
Question 6: L88 - What was the concentration of antimicrobial substances? Also put some detail about the non-treated and solvent control
Response: We apologize for missing this important information. Following your suggestion, we have added the following sentences to ensure calrity.
Antimicrobial agents (0, 12.5, 25 and 50 µg/mL for MEPM, CMZ, and CTRX, 0, 0.1, 1, 10 and 50 µg/mL for LVFX) were then added to the wells and the plates were incubated statically at 37 °C. The 0 µg/mL wells contained an equivalent amount of each solvent. (Lines 100-104)
Question 7: L114 - Please put some information about the non-treated and solvent control
Response: We have added the sentences according to your suggestion.
The wells with 0 µg/mL drug contained an equivalent amount of each solvent and were used as drug-free controls. (Lines 103-104)
Question 8: Figure 1 - Please put information:
- data from controls;
- statistical analysis;
- how did you classify the susceptibility? and what concentrations of drugs did you use?
Response: Thank you for these important suggestions. We have modified the sentences according to your suggestion.
Figure 1 legend
Figure 1. Biofilm-forming capacities and drug susceptibilities of E. coli clinical isolates. Black, gray, and white bars indicate high-capacity (A550 > 0.8), low-capacity (A550 0.1–0.8), and non-biofilm-forming strains (A550 < 0.1), respectively. Measurements were performed three times, and the data shown are the means ± standard deviations. Antimicrobial drug susceptibilities from left to right: meropenem (MEPM), cefmetazole (CMZ), ceftriaxone (CTRX), and levofloxacin (LVFX) of planktonic E. coli cells; the primary diseases are shown next to the strain numbers. AC, acute cholangitis; AOSC, acute obstructive suppurative cholangitis; AP, acute prostatitis; CBDC, common bile duct calculi; Cho, cholecystitis; LA, liver abscess; PS, pyogenic spondylitis; Unk, Unknown; UTI, urinary tract infection. S, susceptible; R, resistant. The drug susceptibilities of planktonic E. coli cells were performed following the Clinical and Laboratory Standards Institute (CLSI) M100-Ed31 protocol.A550 of negative controls without antimicrobial agents was 0.06 (range: 0.055–0.087). (Lines 192-194)
Antimicrobial agents ranges were 1 and 2 µg/mL for MEPM and CTRX, 8, 16, and 32 µg/mL for CMZ, and 0.5, 1. 2 and 4 µg/mL for LVFX. (Lines 98-100)
Question 9: L156 - Please clarify the sentence. I do not understand why did you choose the RB-3 strain? In my opinion RB-56 or RB-50 have better BF parameter and they are sensitive do drug treatment also. However, without statistical analysis and clear information how did you classify the susceptibility/resistanace is difficult to compare.
Response: Of the 52 E. coli strains used, 4 showed high-biofilm-forming capacity. Of these four strains, we selected RB-3 for further analyses as it formed the most stable biofilm.
We have added the following sentences to ensure clarity:
From the four strains exhibiting high biofilm-forming capacity, RB-3 formed the most stable biofilms and was used as a representative for the drug susceptibility testing of biofilms. (Line 201)
Question 10: Figure 2 - Please put the information about control and statistical analysis.
Response: Thank you for highlighting this. Accordingly, we have revised the legend of Figure 2.
Figure 2. Drug susceptibilities of an E. coli RB-3 biofilm to four antibacterial agents. A high-capacity biofilm-forming strain, RB-3, was grown under biofilm-forming conditions for 24 h. Various concentrations of MEPM, CMZ, CTRX (0, 12.5, 25, and 50 μg/mL), and LVFX (0, 0.1, 1.0, 10, and 50 μg/mL) were added followed by incubation at 37 °C for 24 h. The wells without drugs were used as controls. Then the viable cells were counted. The data are the means ± standard deviations of three measurements. MEPM, meropenem; CMZ, cefmetazole; CTRX, ceftriaxone; LVFX, levofloxacin. Statistically significant differences between groups were evaluated using the Kruskal–Wallis test; P < 0.05. The P-value of each figure was less than 0.05. (Lines 212-215)
Question 11: Also why did you use different concentration of LVFX?
Response: Because levofloxacin had a higher antibacterial effect than MEPM, CMZ, and CTRX, dose-response studies were performed at lower concentrations than those of other agents.
Question 12: Figure 3 - Why did you show results only for MEPM treatment?
Response: All of the drugs used in this study are used to treat bacteremia caused by E. coli. Among them, a carbapenem, MEPM, is used when an infection caused by ESBL-producing bacteria is suspected. In recent years, with the increase in cases of infection with ESBL-producing bacteria, it is speculated that there will be more opportunities to administer MEPM empirically. Therefore, in this study, we focused on MEPM in particular.
MEPM and cephalosporin, CMZ and CTRX were similar in susceptibility to biofilms formed by only E. coli, and the sensitivities of these drugs in the presence of C. albicans culture supernatant were similar. These results suggest that the susceptibility of these agents to E. coli–C. albicans dual-species biofilms will be the same as that of MEPM.
We have added the following sentence to clarify this point.
Since the four drugs showed similar behavior against E. coli single biofilm, we tested the effect of MEPM, which can also be used against ESBL-producing bacteria and has high clinical significance, in a dual-species biofilm with C. albicans. (Lines 216-218)
Question 13: Figure 4 - Please change the names: "Medium" and "supernatant" because it is confusing, if I understand properly you treated the E. coli medium with fungal supernatant? But now it looks as you treated the medium and supernatant of E. coli
Response: “Supernatant” in Figure 4 indicates the supernatant of C. albicans. We agree that the term “Supernatant” may confuse the readers. Therefore, we have revised the term “Supernatant” to “Candida culture supernatant (Ca Sup)” according to your suggestion. (Lines 247, 263, 264, 270, 282)

Reviewer 4 Report
In this article by Eshima et al., the authors studied the dual-species biofilm formation by E. coli and C. albicans. They determined the presence of a tolerance-inducing factor from Candida that helped with antibacterial resistance in E. coli present in the biofilm. Although the authors did not identify the factor in this study, they presented data characterizing its nature. The study is well conducted, and the conclusions supported by the data. This article would be of interest to the scientists working on polymicrobial biofilms. There are a few comments and suggestions that would benefit the article.
1. Line 73 – Change to indicate the correct temperature that was used here “at room temperature at 10-30 ℃”
2. Line 47 -…urinary tract or bloodstream respectively.
3. It would be helpful to mention the rationale for choosing the four specific antibacterial agents for the study while discussing the data on the drugs for the first time in the results.
4. Section 2.5 – Reference is missing from line 110. The single species biofilms formed only of Candida is an important control for the entire study. The Candida biofilm formation needs to be described in detail like section 2.2 for E. coli.
5. Section 3.1 – Amongst the ten other biofilm forming E. coli isolates in the table, in contrast to what the authors mention in line 141 and line 142, five of them are also from patients with UTI. It is interesting to note while the antibacterial susceptibility profiles of these bacteria are also like the first four isolates, their biofilm formation capacity is much lower. Was there any other difference between the strains or from the patients' side? Can the authors comment on this observation?
6. Figure 2D – There is an increase in survival of bacteria at 50 µg/ml concentration of levofloxacin compared to the lower concentrations. Does that suggest the generation of drug resistance at this concentration?
7. Can the authors comment on the use of antifungal agents in combination with the antibacterial agents in the treatment of E. coli isolates in the multi species biofilm formation? How does the authors think antifungal- antibacterial combination treatment will work in these situations? The authors’ perspective in this regard (in the discussion) would be of interest to the readers.
8. Candida albicans and E. coli should be italicized throughout the text.
9. Gram staining is named after the bacteriologist who developed the technique and should be written as “Gram” instead of gram. Correct accordingly.
Author Response
# Reviewer 4
In this article by Eshima et al., the authors studied the dual-species biofilm formation by E. coliand C. albicans. They determined the presence of a tolerance-inducing factor from Candida that helped with antibacterial resistance in E. coli present in the biofilm. Although the authors did not identify the factor in this study, they presented data characterizing its nature. The study is well conducted, and the conclusions supported by the data. This article would be of interest to the scientists working on polymicrobial biofilms. There are a few comments and suggestions that would benefit the article.
Response: Dear Reviewer, thank you for your time and efforts in evaluating our manuscript. We are grateful for your insightful suggestions. We agree with your opinions and have revised the manuscript accordingly. We believe the incorporations have further strengthened our manuscript.
Question 1: Line 73 – Change to indicate the correct temperature that was used here “at room temperature at 10-30 ℃”
Response: We have modified the sentence “at room temperature at 10-30 °C” to “at room temperature (10-30°C), according to your suggestion. (Lines 83, 86)
Question 2: Line 47 -…urinary tract or bloodstream respectively.
Response: Thank you for highlighting this. We have modified the sentences as follows according to your suggestion.
… to the urinary tract or bloodstream, respectively [12]. (Line 49)
Question 3: It would be helpful to mention the rationale for choosing the four specific antibacterial agents for the study while discussing the data on the drugs for the first time in the results.
Response: Thank you for this insightful suggestion. We have added the following sentences according to your suggestion.
Antimicrobial susceptibility testing was performed using MEPM, CMZ, CTRX and LVFX against E. coli, the drugs used to treat bacteremia caused by E. coli. (Lines 196-197)
Question 4. Section 2.5 – Reference is missing from line 110. The single species biofilms formed only of Candida is an important control for the entire study. The Candida biofilm formation needs to be described in detail like section 2.2 for E. coli.
Response: Thank you for this important suggestion. Accordingly, we have added the following section to the Materials and methods section of the revised manuscript to describe the Candida biofilm formation method.
Section 2.5 (formerly 2.4)
- C. albicans biofilms were prepared as described above. (Line 137)
2.4. Drug susceptibility testing of biofilms (C. albicans only)
C. albicans SC5314 was grown on Sabouraud dextrose agar at 37 °C for 24 h. Cell suspensions were adjusted to an A630 value of 0.1 in RPMI 1640 medium supplemented with MOPS (pH 7.3). Standardized cell suspensions (100 µL) were seeded into each well of the flat-bottomed 96-well microtiter plates and incubated at 37 °C for 24 h. Planktonic cells were removed, and a medium with 50 μg/mL MEPM was added. The control wells contained an equivalent amount of DMSO. Thereafter, the plates were statically incubated at 37 °C for 24 h, the planktonic cells were removed, and the wells were washed three times with PBS. Viable cells in the biofilms were counted using the CFU method.
(Lines 110-118)
The control wells without antimicrobial agents contained an equivalent amount of DMSO. (Lines 127-128)
The control wells without antimicrobial agents contained an equivalent amount of each solvent. (Lines 140-142)
The wells without MEPM were used as control and contained an equivalent amount of DMSO. (Lines 155-156)
The control wells without supernatant fractions contained an equivalent amount of DMSO. (Lines 164-165)
Question 5. Section 3.1 – Amongst the ten other biofilm forming E. coli isolates in the table, in contrast to what the authors mention in line 141 and line 142, five of them are also from patients with UTI. It is interesting to note while the antibacterial susceptibility profiles of these bacteria are also like the first four isolates, their biofilm formation capacity is much lower. Was there any other difference between the strains or from the patients' side? Can the authors comment on this observation?
Response: Thank you for pointing this out. As you indicated, high and low biofilm-forming strains showed sensitivity to four drugs (with the exemption of RB-08). In the non-biofilm-forming strains, 15 of the 38 strains showed resistance to at least one drug, suggesting a correlation between drug susceptibility and biofilm-formation ability. Hosseini et al. [1] and Leshem et al. [2] reported that the biofilm formation ability of MRSA was higher than that of MSSA. Also, the presence or absence of mecA, a drug-resistant gene in MRSA, affects biofilm phenotype (McCarthy et al. [3]). From this evidence, it is conceivable that the presence or absence of drug-resistant genes may also be involved in biofilm formation in E. coli.
However, the number of strains in this study was limited to 52, so we cannot discuss that possibility. We believe that this is an important issue for future research.
[1]. Hosseini M et al., Microb Drug Resist. 2020 Sep;26(9):1071-1080.,
[2]. Leshem T et al., J Appl Microbiol. 2022 Aug;133(2):922-929.
[3]. McCarthy H et al., Front Cell Infect Microbiol. 2015 Jan 28;5:1.
Question 6. Figure 2D – There is an increase in survival of bacteria at 50 µg/ml concentration of levofloxacin compared to the lower concentrations. Does that suggest the generation of drug resistance at this concentration?
Response: As you pointed out, at 50 µg/mL of LVFX, viable counts were higher at 50 µg/mL concentration than those at 10 µg/mL concentration. Although our study does not provide sufficient reason for this paradoxical effect, a previous study has reported similar paradoxical effects for nalidixic acid, another quinolone. The study speculated that this could be because of the suppression of ROS production at higher concentrations (Luan G et al. [4]). However, the study reported such effects at a high concentration of nalidixic acid (400 µg/mL) and in planktonic cells. We believe that further investigation is necessary to determine whether our results are due to the same reason, “the suppression of ROS production” as nalidixic acid.
[4]. Luan G et al., Antimicrob Agents Chemother. 2018 Feb 23;62(3):e01622-17.
Question 7: Can the authors comment on the use of antifungal agents in combination with the antibacterial agents in the treatment of E. coli isolates in the multi species biofilm formation? How does the authors think antifungal- antibacterial combination treatment will work in these situations? The authors’ perspective in this regard (in the discussion) would be of interest to the readers.
Response: Thank you for this insightful and important suggestion. Our results indicated that C. albicans induced resistance to E. coli biofilm formation. If the biofilm formation of C. albicans can be suppressed, it is possible to reduce drug sensitivity to C. albicans–E. coli dual-species biofilm. However, we cannot deny the possibility that E. coli induces C. albicans to become resistant to antifungal drugs. We have added the following sentences to include this important point in the Discussion section:
Our study implicated that standard treatment may become insufficient for E. coli bacteremia coinfection with C. albicans. In such cases, the combination of antibacterial and antifungal agents effective against C. albicans biofilms may be a better option. (Lines 366-368)
Question 8: Candida albicans and E. coli should be italicized throughout the text.
Response: The italicization of microorganisms’ names has been carefully corrected.
Question 9. Gram staining is named after the bacteriologist who developed the technique and should be written as “Gram” instead of gram. Correct accordingly.
Response: Apologies for this error, We have modified from “gram” to “Gram” in all relevant places in the revised manuscript. (Lines 47, 322, 348, 349)

Reviewer 5 Report
The manuscript by Eshima et al. reports the observation of some increase in resistance to some antimicrobials in E.coli when organized in dual-species biofilm, as well as when exposed to Candida cultures supernatants.
The observation is very interesting and worthy of further investigation, especially from the perspective of improved therapeutic protocols. However, unfortunately, the manuscript is merely descriptive, as it lacks any attempt to identify the underlying mechanism(s) or, at least, hypotheses in this regard.
The authors only tested various treatments of the supernatant, finding that it retains its activity. However, they did not test, among others, the possibility that the simultaneous treatment of dual-species biofilms with anti-fungal and antibacterial drugs could result in restored bacterial sensitivity to the latter. No further attempts to elucidate the possible mechanism(s), including gene expression analyses, are reported in the manuscript.
Author Response
#Reviewer 5
The manuscript by Eshima et al. reports the observation of some increase in resistance to some antimicrobials in E.coli when organized in dual-species biofilm, as well as when exposed to Candida cultures supernatants.
The observation is very interesting and worthy of further investigation, especially from the perspective of improved therapeutic protocols. However, unfortunately, the manuscript is merely descriptive, as it lacks any attempt to identify the underlying mechanism(s) or, at least, hypotheses in this regard.
Response: Dear Reviewer, thank you for your time and efforts in evaluating our manuscript. We are grateful for your insightful suggestions. We agree with your opinions and have revised the manuscript accordingly. We believe the incorporations have further strengthened our manuscript.
Question 1: The authors only tested various treatments of the supernatant, finding that it retains its activity.
Response: As you pointed out, we believe it is necessary to identify the active compounds in the future. We have added the sentences that the purification and structure determination of the active compounds are important issues.
Researchers have shown that farnesol, a small molecule secreted by C. albicans, alters the expression of several genes involved in cell membrane formation and lipid synthesis and impairs the biofilm formation capacities in the Gram-negative bacterium Acinetobacter [22] and Gram-positive bacterium Staphylococcus aureus [28]. Therefore, future studies to purify and characterize the active component (<10 kDa) in Candida culture supernatant using electron microscopy are essential to gain further insights. (Lines 346-347, 348, 349, 349-351)
Question 2: However, they did not test, among others, the possibility that the simultaneous treatment of dual-species biofilms with anti-fungal and antibacterial drugs could result in restored bacterial sensitivity to the latter.
Response: As you pointed out, it is important for future studies to verify whether the combined use of antibacterial and antifungal agents has a therapeutic effect on infections caused by dual-species biofilms through animal experiments. We have added the sentences as follows according to your suggestion.
Although this study revealed that dual-species biofilm of C. albicans and E. coli leads to antimicrobial resistance in E. coli, the effect of C. albicans on antifungal drug resistance and the therapeutic effect of the combination of antimicrobials and antifungal drugs have not been verified. The validation in animal studies of whether the combination of antibacterial and antifungal agents is effective in treating infections caused by dual-species biofilms is an important issue for further investigation. (Lines 368-374)
Question 3: No further attempts to elucidate the possible mechanism(s), including gene expression analyses, are reported in the manuscript.
Response: As you pointed out, we have added our hypothesis and future research development regarding the mechanism of action of drug resistance against E. coli biofilm by C. albicans.
Furthermore, a previous study has shown that C. albicans alters gene expression in P. aeruginosa and affects its iron uptake mechanism [21]. As E. coli also possess an iron-dependent biofilm formation mechanism [33], we assumed that the active compound may have altered gene expression in E. coli by altering the iron-dependent biofilm formation mechanism. Revealing the molecular mechanisms of action of the active compound produced by C. albicans on drug resistance of E. coli will be an important subject. (Lines 351-357)
Reference 33 has been newly added to the text.

Round 2
Reviewer 2 Report
I was expecting that the authors will include more experiments (microscopy in particular) to support their hypothesis by more than one experimental approach, however I think they show satisfactory evidence.
Few points to be considered
1- I would recommend citing this reference in the introduction
El-Telbany M, Mohamed AA, Yahya G, Abdelghafar A, Abdel-Halim MS, Saber S, Alfaleh MA, Mohamed AH, Abdelrahman F, Fathey HA, Ali GH, Abdel-Haleem M. Combination of Meropenem and Zinc Oxide Nanoparticles; Antimicrobial Synergism, Exaggerated Antibiofilm Activity, and Efficient Therapeutic Strategy against Bacterial Keratitis. Antibiotics. 2022; 11(10):1374. https://doi.org/10.3390/antibiotics11101374
2- I would recommend citing this reference in the materials and methods (bacterial sensitivity testing and 16S rRNA sequencing)
A- El-Baz, A. M., Yahya, G., Mansour, B., El-Sokkary, M., Alshaman, R., Alattar, A., & El-Ganiny, A. M. (2021). The Link between Occurrence of Class I Integron and Acquired Aminoglycoside Resistance in Clinical MRSA Isolates. Antibiotics (Basel, Switzerland), 10(5), 488. https://doi.org/10.3390/antibiotics10050488
B- Khalil R, Yahya G, Abdo WS, El-Tanbouly GS, Johar D, Abdel-Halim MS, Eissa H, Magheru C, Saber S, Cavalu S. Emerging Approach for the Application of Hibiscus sabdariffa Extract Ointment in the Superficial Burn Care. Scientia Pharmaceutica. 2022; 90(3):41. https://doi.org/10.3390/scipharm90030041
3- In the section describing culture condition and drug sensitivity of Candida albicans, please cite these references
A- Yahya, G., Hashem Mohamed, N., Pijuan, J., Seleem, N. M., Mosbah, R., Hess, S., Abdelmoaty, A. A., Almeer, R., Abdel-Daim, M. M., Shulaywih Alshaman, H., Juraiby, I., Metwally, K., & Storchova, Z. (2021). Profiling the physiological pitfalls of anti-hepatitis C direct-acting agents in budding yeast. Microbial biotechnology, 14(5), 2199–2213. https://doi.org/10.1111/1751-7915.13904
B- Amponsah, P. S., Yahya, G., Zimmermann, J., Mai, M., Mergel, S., Mühlhaus, T., Storchova, Z., & Morgan, B. (2021). Peroxiredoxins couple metabolism and cell division in an ultradian cycle. Nature chemical biology, 17(4), 477–484. https://doi.org/10.1038/s41589-020-00728-9
C- Yahya, G., Wu, Y., Peplowska, K., Röhrl, J., Soh, Y. M., Bürmann, F., Gruber, S., & Storchova, Z. (2020). Phospho-regulation of the Shugoshin - Condensin interaction at the centromere in budding yeast. PLoS genetics, 16(8), e1008569. https://doi.org/10.1371/journal.pgen.1008569
4- In the section of Fractionation of Candida supernatant, please cite this reference
A- Yahya, G., Parisi, E., Flores, A., Gallego, C., & Aldea, M. (2014). A Whi7-anchored loop controls the G1 Cdk-cyclin complex at start. Molecular cell, 53(1), 115–126. https://doi.org/10.1016/j.molcel.2013.11.015
B- Parisi, E., Yahya, G., Flores, A., & Aldea, M. (2018). Cdc48/p97 segregase is modulated by cyclin-dependent kinase to determine cyclin fate during G1 progression. The EMBO journal, 37(16), e98724. https://doi.org/10.15252/embj.201798724
Reviewer 3 Report
I am satisfied with the author’s responses to my questions/issues raised in my initial review. The revised manuscript is easier to follow based on feedback from the reviewers. I recommend that the revised paper be accepted with very minor revisions. In my opinion in result section it is worth to add information about the statistical analysis results, e.g. the value of F, dF and p.
Reviewer 5 Report
The authors acknowledged the value of the reviewer's comments but failed to address them. They only reported some of those considerations in the revised manuscript, instead of performing experiments.